Review

Subject Area:
biochemistry/cellular biology/molecular biology

Keywords:
hnRNPK, post-translational modifications, phosphorylation, methylation, ubiquitination, glycosylation

Authors for correspondence:
Haixia Xu
e-mail: hxxu214@126.com
Pengpeng Zhang
e-mail: ppzhang15@163.com

# Post-translational modification control of RNA-binding protein hnRNPK function

Yongjie Xu, Wei Wu, Qiu Han, Yaling Wang, Cencen Li, Pengpeng Zhang and Haixia Xu

College of Life Science, Xinyang Normal University, Xinyang 464000, People's Republic of China

HX, 0000-0002-2732-0069

Heterogeneous nuclear ribonucleoprotein K (hnRNPK), a ubiquitously occurring RNA-binding protein (RBP), can interact with numerous nucleic acids and various proteins and is involved in a number of cellular functions including transcription, translation, splicing, chromatin remodelling, etc. Through its abundant biological functions, hnRNPK has been implicated in cellular events including proliferation, differentiation, apoptosis, DNA damage repair and the stress and immune responses. Thus, it is critical to understand the mechanism of hnRNPK regulation and its downstream effects on cancer and other diseases. A number of recent studies have highlighted that several post-translational modifications (PTMs) possibly play an important role in modulating hnRNPK function. Phosphorylation is the most widely occurring PTM in hnRNPK. For example, *in vivo* analyses of sites such as S116 and S284 illustrate the purpose of PTM of hnRNPK in altering its subcellular localization and its ability to bind target nucleic acids or proteins. Other PTMs such as methylation, ubiquitination, sumoylation, glycosylation and proteolytic cleavage are increasingly implicated in the regulation of DNA repair, cellular stresses and tumour growth. In this review, we describe the PTMs that impact upon hnRNPK function on gene expression programmes and different disease states. This knowledge is key in allowing us to better understand the mechanism of hnRNPK regulation.

## 1. Introduction

Protein post-translational modification (PTM) is a process that is a covalent change occurring in almost every protein during or after its translation. PTM follows from various signalling pathways to cause activation of enzymes which play crucial roles in regulating the activity, stability, localization, interactions or folding of proteins by inducing their covalent linkage to new functional chemical groups, such as phosphate, acetyl, methyl, carbohydrate and ubiquitin [1–3]. Different PTM types have different roles for different proteins, such as translocation, secretion, function and elimination. These effects control a group of cells whose proteins are modified by the PTM mechanism, and the response of these cells ranges from activation, survival, proliferation, differentiation and migration to apoptosis [1]. Moreover, PTMs dynamically alter the compartmentalization, trafficking and physical interaction of key molecules that regulate distinct cellular processes. Dysregulated PTMs have been shown to influence disease processes such as cancer, neurodegenerative disorders, virus infection and cardiovascular disease [4].

RNA-binding proteins (RBPs) are typically thought of as proteins that bind RNA through one or multiple globular RNA-binding domains and change the fate or function of the bound RNAs, which play key roles in RNA dynamics, including subcellular localization, alternative splicing, translational efficiency and metabolism [5]. PTMs tightly regulate various functions of the RBPs, including their enzymatic activity, localization and interactions with other proteins, DNA or RNA in response to various physiological and/or pathological

conditions [6]. One of the best studied RBPs with regulatory impact upon a select subset of target RNAs is heterogeneous nuclear ribonucleoprotein K (hnRNPK), which has an abundance of PTMs. PTMs regulate the ability of hnRNPK to control the localization of RNA and splicing changes, as well as regulate translation and transcription. In this review, we summarize and discuss the PTMs of hnRNPK and their impact on hnRNPK function.

## 2. HnRNPK

HnRNPK is the one of the most extensively studied members of an hnRNP family of RBPs that also includes approximately 20 members termed hnRNP A1 through to U [7]; this family of RBPs has been detected in the nucleus, cytoplasm, mitochondria and plasma membrane, interacting with signal transducers, proteins, selective RNA motifs and elements located in the promoter regions of genes [8–10]. HnRNPK is a conserved RNA/DNA-binding protein that is involved in signal transduction and gene expression, and that binds preferentially and tenaciously to transcripts or promoters mainly at poly(C) RNA or DNA stretches to influence their stability and/or translation or transcription. Many genes such as *c-Src*, *c-Myc*, *p21*, *eIF4E*, *r15-LOX* and *UCP2* can serve as hnRNPK targets; these genes encode proteins that take part in cell proliferation, apoptosis and differentiation [11]. Through its impact on the expression of select subsets of proteins, hnRNPK is found to participate in physiological processes such as spermatogenesis, nervous system and ovary development, erythroid and muscle differentiation, organogenesis, responses to stress and carcinogenesis processes [11–13]. Besides coding transcripts, hnRNPK also has the ability to bind to and regulate the functions of miRNAs and long non-coding RNAs (lncRNAs) [14–17].

The relative molecular weight of hnRNPK is approximately 66 kDa, which mainly comprises three K homology domains (KH1, KH2 and KH3), a K-protein interactive region (KI) and a C-terminal protein kinase-binding domain. KH1 and KH2 are separated from KH3 by a segment of approximately 170 amino acids that contains two arginine/glycine/glycine (RGG) boxes [18]. HnRNPK activity is mediated in an incompletely understood way by the three KH domains [9,19,20], and the KH3 domain probably plays the most important binding role, as it has been shown to bind to nucleic acids as an isolated domain, albeit with lower affinity than the full-length protein [19]. Moreover, hnRNPK protein has an N-terminal bipartite nuclear localization signal (NLS) and a nuclear shuttling domain (KNS) located immediately before the C-terminal KH3, probably allowing both nuclear and cytoplasmic functions [21]. Between KH2 and KH3 lies a segment called the KI region that lacks a well-structured three-dimensional fold, which is not found in other poly(C)-binding proteins (PCBPs) [22]. Interestingly, the KI domain, which contains many RGG repeats and proline-rich regions responsible for SH3-domain binding, is evolutionarily conserved between *Xenopus laevis* and mammals but not in fly, nematode and yeast, suggesting that the KI domain emerged later in evolution and allows K-protein to gain new functions such as protein–protein interaction [23]. This domain is responsible for many known hnRNPK protein–protein interactions. For example, the KI region contains three proline-rich motifs that interact with src-homology-3 (SH3) domains, most notably with the SH3 of

the Src family kinases [24–26]. Thus, the modular structure allows hnRNPK to interact with a range of molecular partners and hnRNPK protein is considered as a docking platform that associates signal transduction pathways with nucleic acid-directed processes. Given the distinct roles of the different domains of hnRNPK, PTMs at these regions are important to interact with RNAs or DNA or proteins and directly influence hnRNPK subcellular localization (figure 1). The multiple modifications could respond to a host of signals, allow the hnRNPK protein to sense changes in the extracellular environment and indicate functional diversity of hnRNPK. Here, we review and discuss these modifications of hnRNPK to give us a clear idea of the molecular mechanism of how hnRNPK participates in an abundant biological process. This knowledge will aid in the development of novel therapeutic strategies to target hnRNPK in many types of cancer and other diseases in the future.

## 3. HnRNPK phosphorylation

Among various types of common PTMs, phosphorylation is the most extensively investigated one and plays a crucial role in many distinct biological processes. Phosphorylation modification is also the most profoundly explored modification of hnRNPK protein by mitogens, cytokines and oxidative stress, which impact upon hnRNPK function in different ways, mainly by changing hnRNPK protein stability, affinity for binding nucleic acid or protein and subcellular localization. HnRNPK protein has 31 serine residues, 24 threonine residues and 17 tyrosine residues, together forming 72 potential phosphorylation sites; among these, 20 residues are known to be phosphorylated either *in vivo* or *in vitro* (figure 2) [27]. In general, phosphorylation at or near the hnRNPK KH domains affects hnRNPK binding to target RNA or DNA; for instance, phosphorylation of Y72 in KH1, S189 in KH2, Y458 in KH3 or S216 in the linker region has a minimal influence on hnRNPK structure but affects hnRNPK affinity for RNAs or DNAs [28–30]. By contrast, phosphorylation around the hnRNPK KI or KNS region affects nucleocytoplasmic shuttling, probably by altering the interaction of hnRNPK with the intracellular transport machinery (figure 1). Other phosphorylation events can affect subsequent post-translational modifications, as discussed below. For several specific phosphorylations of hnRNPK, the kinases responsible have been identified. In this section, we review the specific kinases that phosphorylate hnRNPK and their impact on hnRNPK function.

### 3.1. Phosphorylation by MAPKs

Mitogen-activated protein kinases (MAPKs) phosphorylate specific serine and threonine residues of target protein substrates and play a central role in regulating cellular activities ranging from gene expression, mitosis, movement and metabolism to programmed death [31]. There are three well-characterized subfamilies of MAPKs, including extracellular signal-regulated kinases (ERK), c-Jun amino-terminal kinases (JNK) and p38 MAPKs that control a vast array of physiological processes. JNK and ERK, but not p38, effectively phosphorylate hnRNPK protein at serine residues 116, 189, 216, 284 and 353 [29,32–34]. As a target of JNK and ERK kinases, hnRNPK is believed to alter its intracellular distribution or lead to inhibition of mRNA translation, thus it is ideally

royalsocietypublishing.org/journal/rsob Open Biol. 9: 180239

| modifying enzymes | phosphorylating kinases | aa | domains | effect on hnRNPK | reference |
|---|---|---|---|---|---|
| | | | NLS | | |
| | c-Src → Y72 | | KH1 | DNA/RNA binding | 28, 30, 64, 65 |
| | EPK → S116, T120 | | | cytoplasmic export, protein interaction, protein stabilization | 29, 32, 34 |
| | CK2 → S141 | | | unknown | 27 |
| | CK2 → S154 | | | | |
| | ATM → T174 | | KH2 | prevents hnRNPK ubiquitination | 57, 58 |
| | JNK → S189 | | | DNA binding | 29, 32–34 |
| | K198 | | | protein stabilization | |
| HDM2, FBXW7 (ubiquitination) | JNK, CDK2 → S216 | | | DNA/RNA binding | 29, 33, 49, 61 |
| | K219 | | | protein stabilization | 17, 43, 82, 83 |
| | c-Src → Y225, 230, 234, 236 | | | enhances the subsequent phosphorylation | 28, 30, 68, 69 |
| PRMTI (methylation) | R256, R256 | | | location, DNA/RNA banding, interferes with PKCδ-mediated phosphorylation | 72, 75, 77, 78 |
| PRMTl (methylation) | R296, R296 | | KI | | |
| | ERK, CDK2 → S284 | | | cytoplasmic export / location, DNA/RNA binding, nearby site phosphorylation | 29, 32–34, 48, 50 / 72, 75, 77, 78 |
| | PKCδ → S302 | | | translation activation | 21, 52, 54–56 |
| caspase 3 (cleavage) | S334 | | | cleavage | 93 |
| | ERK, JNK → S353 | | KNS | cytoplasmic export, DNA or RNA binding | 29, 32–34, 48, 50 |
| caspase 3 (cleavage) | M359 | | | cleavage | 94 |
| | auror A → S379 | | | transcriptional inhibition | 59 |
| | c-Src → Y380 | | | unknown | 30 |
| | CK2 → S401 | | KH3 | unknown | 27 |
| PIAS3, SENP2 (sumoylation) | K422 | | | DNA repair | 85–88 |
| | ATM → T440 | | | prevents hnRNPK ubiquitination | 57, 58 |
| | c-Src → Y458 | | | RNA binding | 30, 64, 68 |

**Figure 1.** Map of post-translational modifications of hnRNPK. Structure of hnRNPK, depicting three K homology (KH) domains, one N-terminal bipartite nuclear localization signal (NLS), one K-protein interactive region (KI) domain and one nuclear–cytoplasmic shuttling domain (KNS). The modified amino acids are indicated. The modifying enzymes, including the kinases, are indicated on the left. The impact of each modification on hnRNPK localization and function is indicated on the right. References citing the relevant studies are listed.

situated for linking multiple signalling pathways with the gene expressions of numerous cellular processes.

ERK efficiently phosphorylates hnRNPK both *in vitro* and *in vivo* at S284 and S353 [32]. The nuclear–cytoplasmic trafficking of hnRNPK may depend on S284 and S353 phosphorylation, influencing the ability of hnRNPK protein to regulate translation; also, S353 is involved in hnRNPK transactivation activity for increasing the stability of hnRNPK protein-bound transcripts [32,35–38]. Interestingly, S353 is located in the KNS, which is crucial to its intracellular shuttling, and phosphorylated hnRNPK may alter the conformation of the KNS domain. Treatment with the ERK inhibitor PD98059 efficiently blocked hnRNPK cytoplasmic accumulation after serum stimulation or in human erythroid cell maturation [32,37,39,40]. Furthermore, the importance of phosphorylation for cytoplasmic accumulation of hnRNPK protein was further confirmed by the use of hnRNPK protein mutants in which S284 and S353 residues were replaced with acidic residues [32,41]. In differentiated C2C12 cells, we also detected an upregulation of S284 phosphorylation hnRNPK and the cytoplasmic accumulation of hnRNPK protein [42]. In HIV-infected macrophages, the viral-mediated apoptotic block can be released by activating the MAP2K1/ERK2 pathway, which leads to cytoplasmic accumulation of hnRNPK [43]. Similarly, ERK-dependent cytoplasmic accumulation of hnRNPK was upregulated in many human cancers and was involved in the antagonism of cancer cell apoptosis [37,41,44,45]. Activation of ERK results in phosphorylation and cytoplasmic accumulation of hnRNPK, with subsequent translation being regulated by *r15-LOX*, *c-Src*, *UCP2*, *TAK1* and *c-Myc* mRNA [12,39,44,46]. Also, the quantity of hnRNPK in the nucleus decreases, suggesting that the transcriptional activity of genes regulated by hnRNPK

might also be changed. Thus, the ERK-catalysed phosphorylation can also play a role in hnRNPK-bound EGR-1 mRNA transcription splicing, stability and transport to the cytoplasm [47]. Collectively, the ERK-mediated phosphorylation of hnRNPK is important for the function and regulatory mechanism of hnRNPK in various biological processes.

Whereas ERKs are preferentially activated by mitogens, the JNK pathways are triggered primarily by a diverse array of cellular stresses [31]. Thus, the phosphorylation sites for JNK and ERK on the hnRNPK protein are different, and, indeed, JNK phosphorylation results in biological consequences different from those of phosphorylation by ERK [32]. JNK phosphorylation of hnRNPK on S189, S216 and S353 affected neither nuclear export nor concomitant inhibition of RNA translation; however, it increased the transcriptional effects of the hnRNPK protein or mediated the interaction between hnRNPK and the molecular machinery for translating its target RNAs [29,32,48]. In ultraviolet-treated cells, S216 and S353 phosphorylation of hnRNPK increases the AP1-dependent transcriptional activities by promoting its affinity to associated proteins or DNA, which is dependent on JNK activities [29,49]. The increased effect of hnRNPK on transcription is expected to have a wide-ranging effect on transcriptional output owing to the JNK phosphorylation of hnRNPK protein. In addition, S189 phosphorylation of hnRNPK is primarily targeted by JNK in neurons, which is required in developing *X. laevis* neurons for the crucial step in the life cycle of hnRNPK-regulated RNA during axon outgrowth [48,50]. In light of the fact that S189 phosphorylation of hnRNPK occurs within the cytoplasm, ERK may act upstream of the actions of JNK on hnRNPK in regulating translation.

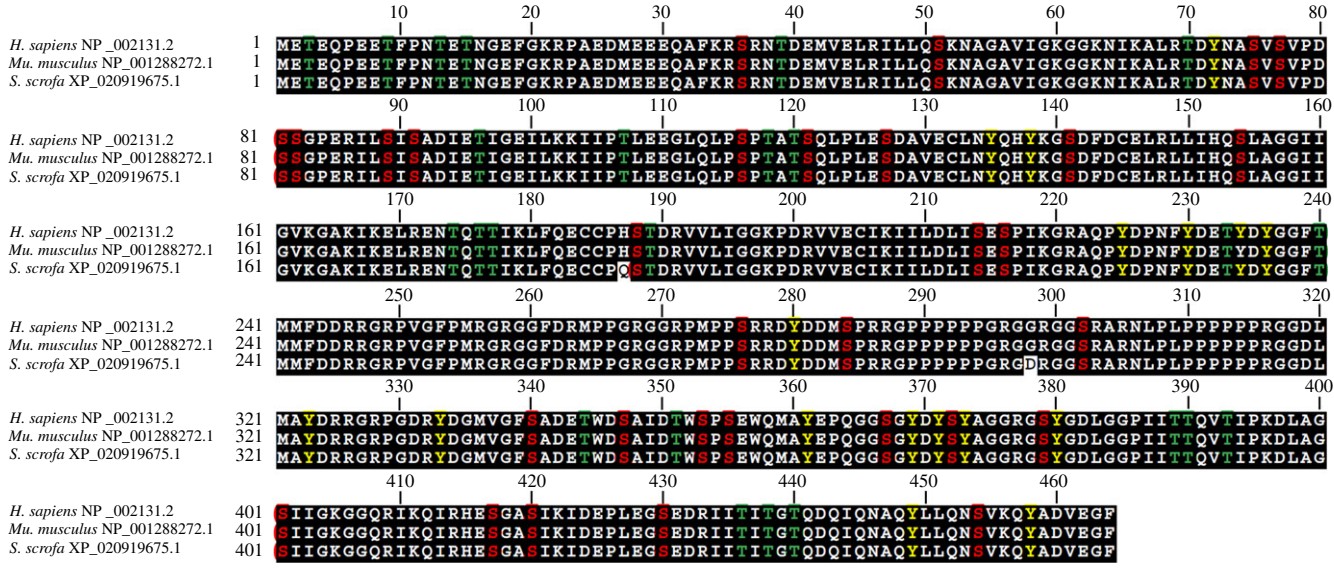

**Figure 2.** The multiple sequences alignment of hnRNPK protein and a detailed map of its defined phosphorylation sites. Red represents a serine phosphorylation site; green represents a threonine phosphorylation site; yellow represents a tyrosine phosphorylation site.

## 3.2. Phosphorylation by PKCδ

Protein kinase C (PKC) consists of a large family of phospholipid-dependent serine/threonine kinases that regulate many different intracellular processes, including cell proliferation, apoptosis, migration, protein secretion and the inflammatory response; the PKC family is divided into three classes based on their $Ca^{2+}$ and lipid requirements [51]. Among the PKC isoenzymes, PKCδ is $Ca^{2+}$-independent but requires phosphatidylserine and diacylglycerol and has unique properties that suggest a functional connection to hnRNPK [52]. *In vivo* studies have shown that hnRNPK and PKCδ remain constitutively bound together within cells, and the target residue S302 for PKCδ-dependent phosphorylation was found in the KI domain of hnRNPK [52–54].

PKCδ efficiently binds and phosphorylates hnRNPK on S302 within the middle of the KI domain, which may serve to modulate the interaction of hnRNPK with its partners, but hnRNPK may also bridge PKCδ to other hnRNPK molecular partners and thus facilitate molecular cross-talk [52,55]. For example, EF-1α binds hnRNPK and is a substrate of PKCδ; the activated PKCδ not only targets S302 on hnRNPK but also phosphorylates EF-1α [21,54]. Therefore, PKCδ-mediated phosphorylation of EF-1α could occur in the context of hnRNPK and contribute to activation of the translational machine. In murine proximal tubular cells, PKCδ-mediated phosphorylation of hnRNPK is required for angiotensin II stimulation of *VEGF* mRNA translation, and hnRNPK phosphorylation correlates with increased *VEGF* mRNA translation and kidney hypertrophy [54,56]. In addition, when DNA is damaged, PKCδ can also be activated by cleaved caspase 3 generated from the apoptosis signal pathway and then enters into the nucleus to phosphorylate hnRNPK, both of which promote apoptosis collaboratively [55].

## 3.3. Phosphorylation by other serine/threonine kinases

HnRNPK can also be phosphorylated by other serine/threonine kinases such as ataxia telangiectasia mutated (ATM), Auror-A, cyclin-dependent kinase 2 (CDK2) and casein kinase-2 (CK2).

In response to ionizing radiation, ATM is activated and phosphorylates hnRNPK on four serine/threonine residues (S121, T174, T370 and T440) [57,58]. Phosphorylation by ATM prevents HDM2-dependent hnRNPK ubiquitination and subsequent degradation, which is required for its stabilization and its function as a p53 transcriptional cofactor in response to DNA damage. Aurora-A, a cell cycle-regulating serine/threonine kinase, was also capable of phosphorylating hnRNPK on S379 in DNA-damaged cells [59]. However, this phosphorylation does not affect the post-transcriptional activity or cellular localization of hnRNPK, but disrupts its interaction with p53 and participates in regulating p53 activity during etoposide-induced DNA damage. Taken together, dynamic phosphorylation of hnRNPK serves as a mechanism for rapidly switching hnRNPK activity and abundance in response to different DNA damage conditions.

Cytosolic accumulation of TAR DNA-binding protein 43 (TDP-43) is a major neuropathological feature of amyotrophic lateral sclerosis and frontotemporal lobar degeneration [60]. Phosphorylation of hnRNPK by CDK2 at S216 and/or S284 involves a key process in the control of TDP-43 cytoplasmic accumulation [61]. CDK2 inhibitors reduced hnRNPK phosphorylation and abrogated accumulation of phosphorylated hnRNPK and TDP-43 during stress. Furthermore, mutation of S216 and S284 phosphorylation sites on hnRNPK inhibited hnRNPK- and TDP-43-positive stress granule formation in transfected cells. Thus, phosphorylation of hnRNPK by CDK2 modulates TDP-43 accumulation and may provide new pharmacological targets for disease intervention.

CK2 is a multi-functional and ubiquitous serine/threonine eukaryotic kinase known to have more than 300 substrates and is involved in signal transduction, transcriptional control, apoptosis, cell cycle regulation and cancer [62]. HnRNPK contains several consensus sites for phosphorylation by CK2, which is capable of being phosphorylated *in vitro* by co-immunoprecipitated CK2 activity from herpes simplex virus type 1 (HSV-1)-infected cells [63]. Phosphorylation of hnRNPK by CK2 could prevent the binding of hnRNPK to RNA, affecting transport of cellular RNAs or altering the subcellular localization of hnRNPK, which could play a key role in HSV-1

infection such as by targeting IE63 to transcriptionally active nuclear domains or facilitating its access to one of the molecular partners of hnRNPK such as DNA or another cellular protein. Furthermore, Mikula *et al.* [27] reported that 17 phosphorylated sites of hnRNPK were identified by mass spectrometry *in vitro* and *in vivo* by CK2. The results indicated that the phosphorylation of hnRNPK by CK2 is complicated. However, the functional mechanism of most sites for phosphorylation by CK2 was still unclear, and the mass spectrometry analysis does not distinguish whether or not a given set of phosphopeptide digests came from the same molecule or different molecules of hnRNPK protein. Thus, whether hnRNPK protein exists in a limited or a very large number of CK2 phosphorylation states and the function of the phosphorylation site requires further research.

## 3.4. Phosphorylation by c-Src

All of the phosphorylated residues on hnRNPK discussed thus far are serine or threonine. However, tyrosine phosphorylation of hnRNPK is also emerging as important for regulating hnRNPK function. The best characterized mechanism for tyrosine phosphorylation of hnRNPK is that mediated by the non-receptor tyrosine kinases c-Src. This tyrosine kinase can directly bind to hnRNPK, leading to phosphorylation on seven tyrosine residues including Y72, Y225, Y230, Y234, Y236, Y380 and Y458 out of the 17 tyrosine residues present in hnRNPK [28,30,64,65]. Y72 and Y458, respectively, are located at the KH1 and KH3 domains of hnRNPK. Phosphorylation of these residues affects their nucleic acid-binding affinity, reducing hnRNPK interaction with targets such as *r15-LOX*, *TAK1*, *UCP2* and *NMHC IIA* mRNA [27,28,46,66,67]. For example, *r15-LOX* mRNA translation activation is necessary for mature reticulocytes, while phosphorylation of Y458 in the KH3 domain diminishes the differentiation control element (DICE) binding activity of hnRNPK; consequently, its function as an inhibitor of *r15-LOX* mRNA translation is lost [30,64,68]. The phosphorylation of the other five tyrosine residues showed different functions from Y72 and Y458, which are located in the linker region of hnRNPK. For example, in the case of oxidative stress, phosphorylation of Y230, Y234, Y236 or a sequential combination of these tyrosine residues could be the priming phosphorylation event. The priming role of Y230, Y234 or Y236 is supported by the observation that *in vivo* phosphorylation of these tyrosine residues seems to occur rapidly and by the fact that tyrosine phosphorylation of hnRNPK protein greatly enhances the subsequent phosphorylation of S302 by PKCδ *in vitro* and *in vivo* [68,69]. Surprisingly, however, hnRNPK has been shown not only to be phosphorylated by c-Src kinases but also to activate the kinases in cell culture and in mouse liver in response to oxidative stress.

## 4. HnRNPK methylation

Protein arginine methylation is one of the most frequent PTMs in mammals and involves the regulation of biological processes in eukaryotes, such as RNA splicing, protein interactions, nuclear/cytoplasmic transport, transcription and translation [70]. Protein arginine methyl transferases (PRMTs) catalyse the transfer of a methyl group from *S*-adenosylmethionine to specific arginine residues of a target protein generating monomethylated and dimethylated arginine. Protein arginine

methyltransferase 1 (PRMT1) generates the majority of asymmetric dimethylarginine residues [71].

HnRNPK is a key target of PRMT1, which contains some arginine-rich regions which are typically asymmetrically dimethylated at the Arg–Gly–Gly motif (RGG box) or RG repeat regions [23,72–74]. Mass spectrometry analysis has identified five major (R256, R258, R268, R296 and R299) and two minor (R303 and R287) methylarginines in the RGG motif of hnRNPK [72,75]. Among these methylation sites, R296 and R299 show higher methylation activity, suggesting that these two sites may play distinct roles in the biological function of hnRNPK. PRMT1-mediated arginine methylation does not influence the RNA-binding activity of hnRNPK or its translation inhibitory function but changes hnRNPK function in regulating its interaction with p53, c-Src and Epstein–Barr virus nuclear antigen 2 (EBNA2) proteins [23,73,76], influencing its intracellular distribution [72] or inhibiting its nearby modifications [77]. HnRNPK methylation in erythroid differentiation suppresses the interaction of its proline-rich domains with the SH3 domain of c-Src, and inhibition of methylation can result in an active state that catalyses tyrosine phosphorylation of hnRNPK [23,78]. Arginine methylation of hnRNPK by PRMT1 is likely to increase the affinity for p53 or EBNA2, and to play a key role in stimulating the p53 transcriptional activity required for the DNA damage response or in promoting the EBNA2-dependent activation of the viral latent membrane protein 2A promoter [73,76,79]. Another study showed that methylation by PRMT1 could strengthen the association of hnRNPK in nuclear retention in HEK293 cells and is one of the multiple mechanisms regulating nuclear localization of hnRNPK [72]. Methylated hnRNPK also correlated significantly with inhibiting nearby PKCδ-mediated phosphorylation of hnRNPK to suppress the PKCδ-mediated apoptosis signalling pathway [77]. It is because methylation of R296 and R299 can give rise to space inhibition effects between PKCδ and hnRNPK protein that PKCδ-mediated phosphorylation of S302 on hnRNPK is affected both *in vitro* and *in vivo*.

## 5. HnRNPK ubiquitination

Protein ubiquitination is inversely regulated by E1 ubiquitin-activating enzyme, E2 ubiquitin-conjugating enzyme and E3 ubiquitin ligase or de-ubiquitination enzymes. Protein ubiquitination plays a critical role in a plethora of cellular processes such as proteasomal degradation, autophagy, assembly of multi-protein complexes, inflammatory signalling, cell cycle progression, cell differentiation, immune defence, DNA repair and regulation of enzymatic activity [80,81]. HnRNPK plays key roles in coordinating transcriptional responses to DNA damage, and induction of hnRNPK ensues through the inhibition of its ubiquitin-dependent proteasomal degradation mediated by the ubiquitin E3 ligase HDM2/MDM2 [17,43,82]. Upon DNA damage stress, ubiquitination-mediated low expression of p53 and hnRNPK will de-ubiquitinate to improve their stability and protein levels and then cooperate with p53 in transcriptional activation of cell cycle-related and pro-apoptotic genes in repair processes [82]. When the repair is completed, hnRNPK will revert to maintain a low level once again through the proteasome pathway and cells consequently enter into a normal lifecycle. Furthermore, the tumour suppressor FBXW7 is a part of the SCF (complex of SKP1, Cullin 1, F-box protein) ubiquitin ligase complex, which also controls the degradation

royalsocietypublishing.org/journal/rsob   *Open Biol.* **9**: 180239

of hnRNPK protein by the proteasome pathway [83]. In patients with pancreatic ductal adenocarcinoma cancer, the inhibition of FBXW7-mediated ubiquitination/proteasome activity was found to elevate hnRNPK expression, but phosphorylation by GSK3$\beta$ and ERK induced the proteasomal degradation of hnRNPK. Taken together, dynamic ubiquitination and de-ubiquitination of hnRNPK serves as a mechanism for rapidly switching hnRNPK activity and abundance in response to cellular stresses.

## 6. HnRNPK sumoylation

Similar to ubiquitination, sumoylation is a multi-step reaction that covalently conjugates a 12-kDa small ubiquitin-like modifier (SUMO) to a variety of cellular proteins by a cascade enzyme system consisting of a single E1-activating enzyme, a unique E2-conjugating enzyme (Ubc9) and different E3 ligases [84]. Sumoylation plays an important role in many cellular processes, and SUMO-targeted proteins are being identified at an accelerating rate. In the human embryonic kidney 293 Tet-On cell line, hnRNPK K422 sumoylation was first identified using *in vivo* and *in vitro* methods to enrich the amounts of sumoylated proteins and affinity purify them to facilitate the mass spectrometry identification of SUMO target proteins [85]. Sumoylated hnRNPK isolated from cell lines stably expressing SUMO supports the notion that hnRNPK sumoylation may play a role in the regulation of RNA metabolism at the transcriptional as well as post-transcriptional levels. Recently, Suk *et al.* [86] found that sumoylated hnRNPK positively regulates c-Myc expression at the translational level, which contributes to Burkitt's lymphoma cell proliferation. Another study showed that hnRNPK is modified by SUMO in K422 and sumoylation is regulated by the E3 ligase Pc2/CBX4 [87]. Moreover, DNA damage stimulates hnRNPK sumoylation through Pc2 E3 activity, and this modification is required for p53 transcriptional activation. These findings linked DNA damage-induced Pc2 activation to p53 transcriptional co-activation through hnRNPK sumoylation. In addition, another study reported that ultraviolet radiation induces E3 ligase PIAS3-mediated sumoylation of hnRNPK, which increases hnRNPK stability, interaction between hnRNPK and p53, and p21 expression in an ATM-Rad3-related (ATR)-dependent manner, leading to cell cycle arrest [88]. In the same report, when the restoration is about to complete, SENP2 competes with PIAS3 to bind to hnRNPK and removes SUMO from hnRNPK, causing the complex to rapidly dissociate into a single component and cells are returned to normal growth after DNA repair. Therefore, in response to DNA damage, reversible sumoylation of hnRNPK by PIAS3 and SENP2 plays a crucial role in the control of hnRNPK stability and appropriate conditions to promote DNA repair and cell cycle recovery. Taken together, dynamic sumoylation and desumoylation of hnRNPK serves as a mechanism for rapidly switching hnRNPK activity and abundance in response to different DNA damage conditions.

## 7. HnRNPK glycosylation

Glycosylation, as the most common PTM in cells, is the attachment of a carbohydrate to a hydroxyl or other functional group of amino acids and is known to play a role in establishing tissue architecture, transmitting signals across the plasma membrane and tumorigenesis [89]. O-GlcNAcylation is one type of glycosylation mediated by O-linked *N*-acetylglucosaminyltransferase. The reversible O-GlcNAcylation modification of serine or threonine residues of numerous proteins constitutes an important glucose-sensing signalling pathway [90]. Recently, Phoomak *et al.* [91] reported the interesting result that O-GlcNAcylation modification of hnRNPK is implicated in the mediation of nuclear translocation in addition to the migration of cholangiocarcinoma (CCA) cells. Moreover, nuclear expression of hnRNPK is positively correlated with high O-GlcNAcylation levels, metastatic stage and shorter survival of patients with CCA. Thus, hnRNPK O-GlcNAcylation may act as a promising therapeutic target to suppress CCA progression. However, further studies are needed to fully understand the precise role of O-GlcNAcylation on nuclear translocation of hnRNPK using the site-directed mutagenesis of O-GlcNAcylation on hnRNPK.

## 8. Proteolytic cleavage

Caspases not only catalyse site-specific protein cleavage in apoptosis but also exert subtle non-apoptotic functions in the differentiation of several myeloid lineages [92]. Upon terminal erythroid differentiation, caspase-3-catalysed hnRNPK cleavage at aspartate (D)334 and glycine (G)335 leads to the generation of a smaller inactive hnRNPK product, the DICE-binding KH domain 3 from the N-terminal part [93]. The RNA-binding activity of hnRNPK is conferred by the KH3 domain that interacts with the 3′-UTR DICE of the *r15-LOX* mRNA, which is critical for its function in translational regulation in erythroid differentiation. During late erythroid cell maturation, the degradation of mitochondria is initiated by reticulocyte r15-LOX, and caspase-3-catalysed hnRNPK KH3 cleavage provides a save–lock mechanism for timely r15-LOX synthesis activation [93]. Recently, it was reported that Granzyme M cleaves hnRNPK at multiple sites and that hnRNPK is required for efficient human cytomegalovirus (HCMV) replication by promoting cell viability in virus-infected cells [94]. The fragment of hnRNPK that is generated by Granzyme M in infected cells contributes to the mechanism by which cytotoxic lymphocytes inhibit HCMV replication, which has a similar molecular weight to the caspase 3 cleavage product in erythroid differentiation, but cleavage at amino acids D334 was not detected; instead, M359 was mapped as the cleavage site. In sum, proteolytic hnRNPK cleavage by caspases or Granzyme M plays a role in vital cellular processes such as erythroid maturation and inhibits HCMV replication. HnRNPK cleavage products may influence the post-transcriptional fate of target mRNAs, but the entire spectrum of actions resulting from hnRNPK cleavage remains to be studied.

## 9. Concluding remarks and perspectives

PTMs of hnRNPK are involved in important biological processes including carcinogenesis, apoptosis, tumorigenesis and cancer progression, spermatogenesis, axonal regeneration, erythroid differentiation, myogenesis, cell division and cell response to stress agents. HnRNPK function is regulated via changes in its subcellular localization, affinity for RNA or DNA, interaction with protein, abundance and cleavage, which in turn influence hnRNPK's ability to affect the fate of

royalsocietypublishing.org/journal/rsob  *Open Biol.* **9**: 180239

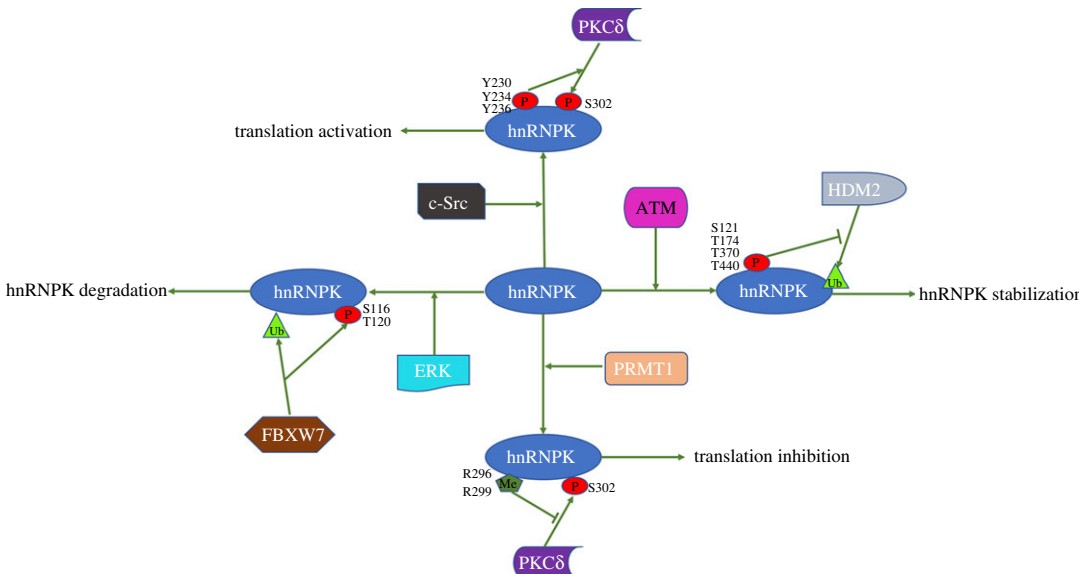

**Figure 3.** This is a brief summary about several PTMs of hnRNPK protein appear to coordinate to elicit a specific outcome. The left side of the figure shows that ERK phosphorylation of hnRNPK at S116 and T120 leads to ubiquitin-mediated degradation of hnRNPK; the right side of the figure shows that ATM-mediated phosphorylation at T174 and T440 can block ubiquitin-mediated degradation of hnRNPK. The figure shows that c-Src phosphorylation of hnRNPK at Y230, Y234 and Y236 enhances the PKCδ-mediated phosphorylation of S302 on hnRNPK; the chart below shows that PRMT1-mediated methylation at R296 and R299 can block the PKCδ-mediated phosphorylation of S302 on hnRNPK. P, phosphorylation; Me, methylation; Ub, ubiquitination.

target nucleic acids. Thus, PTMs enable hnRNPK to elicit quick changes in gene expression programmes.

Several studies have reported hnRNPK phosphorylation by different kinases with various impacts on gene expression and cell fates. ERK, JNK, PKCδ and c-Src are major kinases that phosphorylate hnRNPK and modulate cell proliferation and survival. Phosphorylation by ERK changes the subcellular distribution of hnRNPK and subsequent binding to target transcripts. Phosphorylation by JNK increases the transcriptional effects of the hnRNPK protein or mediates the interaction between hnRNPK and the molecular machinery for translating its targeted RNAs. Phosphorylation by PKCδ under DNA damage conditions influences hnRNPK binding to target mRNAs without altering its subcellular localization. Phosphorylation by c-Src affects its nucleic acid-binding affinity, reducing hnRNPK interaction with targets. Although phosphorylation at or near KHs generally affects hnRNPK binding to target RNA/DNA, hnRNPK phosphorylation at the same residue by different kinases may have distinct outcomes; for example, phosphorylation at S353 by JNK influences hnRNPK binding to proteins or DNA, increasing the AP1-dependent transcriptional activities [29], while S353 phosphorylation by ERK causes hnRNPK to accumulate in the cytoplasm and subsequently regulate mRNA translation [32]. Additionally, phosphorylation events that affect hnRNPK subcellular localization in a similar manner may have distinct outcomes for hnRNPK binding to target nucleic acids or proteins. For example, hnRNPK phosphorylation by ERK increases hnRNPK in the cytoplasm and enhances its interaction with *r15-LOX* mRNA [39], while hnRNPK phosphorylation by CDK2 also elevates cytoplasmic hnRNPK levels, promoting hnRNPK binding to TDP-43 protein [60]. These examples suggest that the PTMs of hnRNPK cannot be interpreted in isolation and instead may involve other factors such as the particular stimulus, the cellular environment and perhaps other concurrent PTMs of hnRNPK. Regarding the last possibility, several PTMs appear to coordinate to

elicit a specific outcome (figure 3); for instance, ubiquitination and de-ubiquitination can alter hnRNPK levels and activity dynamically depending on the stimulus and the cellular response. While ERK phosphorylation of hnRNPK at S116 and T120 leads to ubiquitin-mediated degradation of hnRNPK [83], ATM-mediated phosphorylation at T174 and T440 can block it [58].

Despite considerable progress on the PTMs that affect hnRNPK function in determining the multi-functional roles of hnRNPK in regulating cellular homeostasis, there are still many important questions that need to be addressed, as follows. How do they impact upon the affinity of hnRNPK for target RNAs or DNAs? How do they modify the interaction of hnRNPK with other proteins? How do the different PTMs jointly regulate hnRNPK abundance, subcellular localization and activity? Cells may also regulate hnRNPK through a variety of protein kinases and signalling pathways. What signalling pathways regulate the effectors of hnRNPK PTM? How these impinge on hnRNPK is unclear but they could provide important clues regarding the regulation of hnRNPK functions. What processes and proteins participate in the reversion of these PTMs (e.g. phosphatases, demethylases and de-ubiquitinating enzymes)? Answers to these questions await a comprehensive integration of genetic, biochemical, molecular biology and structural analyses focused on hnRNPK protein–protein and protein–RNA complexes and will highlight novel targets for therapeutic intervention.

**Data accessibility.** This article has no additional data.

**Competing interests.** The authors declare no financial or nonfinancial competing interests.

**Funding.** This work was financially supported by the National Natural Sciences Foundation of China (U1204326 and 31601167), the National Natural Sciences Foundation of Henan Province (182300410027), the Nanhu Scholars Program of XYNU, the Program of Youth Learning Backbone Teacher in Henan province (2015GGJS-139) and the Scientific Research Foundation of Graduate School of Xinyang Normal University (2018KYJJ16).

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
