## [Reviewer comments · Open Biology]

Review History

RSOB-18-0239.R0 (Original submission)

Review form: Reviewer 1

Recommendation

Accept with minor revision (please list in comments)

Are each of the following suitable for general readers?

- a) **Title**
Yes
- b) **Summary**
Yes
- c) **Introduction**
Yes

Is the length of the paper justified?

No

Should the paper be seen by a specialist statistical reviewer?

No

Is it clear how to make all supporting data available?

Not Applicable

Is the supplementary material necessary; and if so is it adequate and clear?

Not Applicable

Do you have any ethical concerns with this paper?

No

Comments to the Author

The review showed different aspects of posttranslational modifications in hnRNPK protein and described their impact in hnRNPK function. I have some comments:

1- the abstract could indicate the alterations associated to the modifications described in hnRNPK in the present manuscript more than describe hnRNPK protein itself;

2- In my opinion the "1.Introduction" should be excluded and other general informations in all manuscript too.

3- In lines 94-96, the authors did a statement based on their review "a clear idea of the molecular mechanism of how hnRNPK participating in abundant biological process". However, I suggest to modify it. What is the motivation to write the revision and what the authors can conclude after that? What are the questions unknown in PTM for hnRNPK? Do you think that PTM is the most important mechanism to control hnRNP K function?

Minor points:

1- The authors should correct typos and English.

2- Fig.1- check the last effect listed in the figure (right side) "banding RNA", it's correct?

3-Fig.2- What is the hnRNPK isoform used to perform the alignment? Include the access number.

Decision letter (RSOB-18-0239.R0)

28-Jan-2019

Dear Dr Xu,

We are pleased to inform you that your manuscript RSOB-18-0239 entitled "Post-translational modification control of RNA-binding protein hnRNPK function" has been accepted by the Editor for publication in Open Biology. The reviewer has recommended publication, but also suggest some minor revisions to your manuscript. Therefore, we invite you to respond to the reviewer's comments and revise your manuscript.

Please submit the revised version of your manuscript within 14 days. If you do not think you will be able to meet this date please let us know immediately and we can extend this deadline for you.

To revise your manuscript, log into <https://mc.manuscriptcentral.com/rsob> and enter your Author Centre, where you will find your manuscript title listed under "Manuscripts with

Decisions." Under "Actions," click on "Create a Revision." Your manuscript number has been appended to denote a revision.

When submitting your revised manuscript, you will be able to respond to the comments made by the referee(s) and upload a file "Response to Referees" in "Section 6 - File Upload". You can use this to document any changes you make to the original manuscript. In order to expedite the processing of the revised manuscript, please be as specific as possible in your response to the referee.

- 1) A text file of the manuscript (doc, txt, rtf or tex), including the references, tables (including captions) and figure captions. Please remove any tracked changes from the text before submission. PDF files are not an accepted format for the "Main Document".
- 2) A separate electronic file of each figure (tiff, EPS or print-quality PDF preferred). The format should be produced directly from original creation package, or original software format. Please note that PowerPoint files are not accepted.
- 3) Electronic supplementary material: this should be contained in a separate file from the main text and meet our ESM criteria (see <http://royalsocietypublishing.org/instructions-authors#question5>). All supplementary materials accompanying an accepted article will be treated as in their final form. They will be published alongside the paper on the journal website and posted on the online figshare repository. Files on figshare will be made available approximately one week before the accompanying article so that the supplementary material can be attributed a unique DOI.

Online supplementary material will also carry the title and description provided during submission, so please ensure these are accurate and informative. Note that the Royal Society will not edit or typeset supplementary material and it will be hosted as provided. Please ensure that the supplementary material includes the paper details (authors, title, journal name, article DOI). Your article DOI will be 10.1098/rsob.2016[last 4 digits of e.g. 10.1098/rsob.20160049].

- 4) A media summary: a short non-technical summary (up to 100 words) of the key findings/importance of your manuscript. Please try to write in simple English, avoid jargon, explain the importance of the topic, outline the main implications and describe why this topic is newsworthy.

Images

Data-Sharing

It is a condition of publication that data supporting your paper are made available. Data should be made available either in the electronic supplementary material or through an appropriate repository. Details of how to access data should be included in your paper. Please see <http://royalsocietypublishing.org/site/authors/policy.xhtml#question6> for more details.

Sincerely,
The Open Biology Team
mailto:openbiology@royalsociety.org

Reviewer's Comments to Author:

Referee:

Comments to the Author(s)

The review showed different aspects of posttranslational modifications in hnRNPk protein and described their impact in hnRNPk function. I have some comments:

1- the abstract could indicate the alterations associated to the modifications described in hnRNPk in the present manuscript more than describe hnRNPk protein itself;

2- In my opinion the "1.Introduction" should be excluded and other general information in all the manuscript too.

3- In lines 94-96, the authors did a statement based on their review "a clear idea of the molecular mechanism of how hnRNPk participating in abundant biological process". However, I suggest to modify it. What is the motivation to write the revision and what the authors can conclude after that? What are the questions unknown in PTM for hnRNPk? Do you think that PTM is the most important mechanism to control hnRNPk function?

Minor points:

1- The authors should correct typos and English.

2- Fig.1- check the last effect listed in the figure (right side) "banding RNA", it's correct?

3- Fig.2- What is the hnRNPk isoform used to perform the alignment? Include the access number.

Decision letter (RSOB-18-0239.R1)

01-Feb-2019

Dear Dr Xu

We are pleased to inform you that your manuscript entitled "Post-translational modification control of RNA-binding protein hnRNPk function" has been accepted by the Editor for publication in Open Biology.

Sincerely,

The Open Biology Team
mailto: openbiology@royalsociety.org